# Peripherally administered orexin improves survival of mice with endotoxin shock

**Yasuhiro Ogawa[1†], Yoko Irukayama-Tomobe[1†], Nobuyuki Murakoshi[2], Maiko Kiyama[1], Yui Ishikawa[1], Naoto Hosokawa[1], Hiromu Tominaga[1], Shuntaro Uchida[1], Saki Kimura[3], Mika Kanuka[1], Miho Morita[1], Michito Hamada[3], Satoru Takahashi[3], Yu Hayashi[1], Masashi Yanagisawa[1,4]\***

[1]International Institute for Integrative Sleep Medicine (WPI-IIIS), University of Tsukuba, Tsukuba, Japan; [2]Department of Cardiology, Faculty of Medicine, University of Tsukuba, Tsukuba, Japan; [3]Department of Anatomy and Embryology, Faculty of Medicine, University of Tsukuba, Tsukuba, Japan; [4]Department of Molecular Genetics and Howard Hughes Medical Institute, Unversity of Texas Southwestern Medical Center, Dallas, United States

**Abstract** Sepsis is a systemic inflammatory response to infection, accounting for the most common cause of death in intensive care units. Here, we report that peripheral administration of the hypothalamic neuropeptide orexin improves the survival of mice with lipopolysaccharide (LPS) induced endotoxin shock, a well-studied septic shock model. The effect is accompanied by a suppression of excessive cytokine production and an increase of catecholamines and corticosterone. We found that peripherally administered orexin penetrates the blood-brain barrier under endotoxin shock, and that central administration of orexin also suppresses the cytokine production and improves the survival, indicating orexin's direct action in the central nervous system (CNS). Orexin helps restore body temperature and potentiates cardiovascular function in LPS-injected mice. Pleiotropic modulation of inflammatory response by orexin through the CNS may constitute a novel therapeutic approach for septic shock.

**\*For correspondence:** yanagisawa.masa.fu@u.tsukuba.ac.jp

[†]These authors contributed equally to this work

**Competing interests:** The authors declare that no competing interests exist.

## Introduction

Systemic inflammatory response syndrome induced by infection, or sepsis (*Bone et al., 1992*; *Dellinger et al., 2013*), can lead to a life-threatening medical emergency requiring intensive care (*Martin et al., 2003*). Septic shock is defined as cardiovascular dysfunction triggered by sepsis, representing the most severe stage of the illness (*Angus and van der Poll, 2013*). Lipopolysaccharide (LPS), a major cell wall component of Gram negative bacteria, plays a central role in sepsis as the endotoxin inducing a systemic inflammatory response, and LPS-induced endotoxin shock is one of the several well-studied animal models of septic shock. Recent advances have started to reveal the highly complex pathophysiology of sepsis (*Cohen, 2002*). However, researchers have failed to translate the advances in understanding the pathophysiology of sepsis into effective new therapies, and the mortality of sepsis still remains high. To improve the outcome of patients with sepsis, new therapeutic strategies and agents are essential.

Recent studies have revealed the regulation and integration of inflammatory responses by the central nervous system (CNS) through the neuroendocrine and autonomic nervous systems (*Tracey, 2002*). For example, vagal afferents activated by endotoxin and cytokines in sepsis stimulate the hypothalamic-pituitary-adrenal axis and exert anti-inflammatory effects through the release

**eLife digest** The body has a range of defenses to fight infection, which play a crucial role in keeping us healthy. However, sometimes the response to infection may damage the body's own tissues and organs, leading to a life-threatening condition called sepsis. In the most severe stage of sepsis – known as septic shock – blood pressure drops to dangerously low levels and the individual often dies.

There is currently no effective therapy for septic shock. Recent studies have revealed how the brain regulates immune responses via chemical signals and nerve impulses. A molecule called orexin is made in the brain and regulates the activity of a group of neurons that control sleep. Orexin can also alter heart rate and body temperature in rats, which suggests that it may have potential to be developed as a treatment for septic shock.

To test this idea, Ogawa, Irukayama-Tomobe et al. injected orexin under the skin of mice with septic shock. The experiments show that the injected orexin is able to enter the brain, where it helps the mice to survive and recover from septic shock by restoring normal body temperature and boosting heart rate. Further experiments suggest that orexin is likely to regulate immune responses through multiple signaling pathways in the brain.

The next step following on from this work is to find out the precise mechanism through which orexin regulates the responses of the immune system. This orexin treatment strategy should also be tested on primates with septic shock before planning any clinical trials in humans.

of glucocorticoids (*Tracey, 2002*). A cholinergic anti-inflammatory pathway has also been reported, in which the activation of efferent vagus nerves suppresses systemic inflammatory responses (*Borovikova et al., 2000*; *Wang et al., 2004*). Acetylcholine attenuates cytokine production from LPS-activated macrophages in the spleen through the nicotinic acetylcholine receptor (*Wang et al., 2003*). Vagus nerve stimulation also leads to the activation of the splenic nerve and release of norepinephrine in the spleen (*Vida et al., 2011*). Norepinephrine inhibits cytokine production in the spleen and suppresses systemic inflammation in experimental sepsis through $\beta2$-adnenoceptors on lymphocytes (*Vida et al., 2011*). Thus, the anti-inflammatory effects of the sympathetic and parasympathetic nervous systems seem to be synergistic.

The hypothalamic neuropeptide orexin, which plays a crucial role in controlling sleep/wakefulness (*Sakurai, 2007*), regulates the hypothalamo-pituitary-adrenal axis by activating the paraventricular nucleus (PVN) (*Kuru et al., 2000*) and integrates autonomic functions by interacting with brainstem centers (*Zheng et al., 2005*). Recent reports show that intracerebroventricular (ICV) administration of orexin modulates heart rate and body temperature, and increases the level of adrenocorticotropic hormone (ACTH) in a murine sepsis model induced by cecal ligation and puncture (*Deutschman et al., 2013*). ICV orexin also partially increases locomotor activity suppressed by a low dose of LPS in rats (*Grossberg et al., 2011*). However, no information is available as to whether orexin actually improves the survival and/or suppresses the systemic inflammation. Moreover, CNS administration of therapeutic agents in human patients may often be unfeasible; clinical applications of orexin in humans require a tactic to deliver orexin into the brain. Systemic inflammation, the hallmark of sepsis, enhances the permeability of the blood-brain barrier (BBB) in rodents (*Kowal et al., 2004*; *Xaio et al., 2001*) and humans (*Ballabh et al., 2004*); some peptides and proteins, including insulin, albumin (*Xaio et al., 2001*), and antibodies (*Kowal et al., 2004*), can enter the brain under the condition of systemic inflammation. In this study, we deliver orexin into the brain by taking advantage of the enhanced BBB permeability under the condition of systemic inflammation, rescuing mice with endotoxin shock by targeting the CNS.

# Results

## Peripherally administered orexin improves survival of mice with endotoxin shock

Orexin cannot normally penetrate the BBB (*Fujiki et al., 2003*). Surprisingly, however, we found that the survival of mice with endotoxin shock was dramatically improved (*Figure 1A*) by subcutaneous (SC) infusion of orexin-A (OXA; 1 mg/mouse/24 hr) starting 30 min before a lethal dose of LPS injection (10 mg/kg IP, which kills 70–90% of mice). No effect was seen in $Hcrtr1^{-/-}$;$Hcrtr2^{-/-}$ mice (abbreviated as OXRKO mice) (*Figure 1B*). Furthermore, by SC administration of OXA (2 mg/mouse/24 hr) starting 30 min after LPS injection, the survival rate and duration were also significantly improved (*Figure 1C*). Although orexin was administered for the first 24 hr only, its life-supporting effect manifested over the next several days.

## Orexin inhibits cytokine production in mice with endotoxin shock

LPS induces excessive cytokine production systemically, leading to further amplification of the inflammatory response (*Beutler and Rietschel, 2003*). We measured the levels of 32 cytokines in serum from LPS-injected mice. Whereas LPS elevated the levels of most cytokines, SC infusion of OXA (1 mg/mouse/24 hr) significantly inhibited the increase of a number of cytokines at 4 hr and 22 hr after LPS injection (*Figure 2*). Simultaneously, OXA ameliorated hypothermia induced by LPS at 4 hr and 22 hr after LPS injection (*Figure 2—figure supplement 1*). Particularly, in the early stage of endotoxin shock, the levels of cytokines produced by LPS-activated macrophages (e.g., tumor necrosis factor-α (TNF-α), macrophage inflammatory protein-1α (MIP-1α, also termed CCL3), and MIP-1$\beta$ [CCL4]) were significantly decreased by OXA treatment. In the later stage, the levels of many more cytokines produced by macrophages and/or lymphocytes were significantly decreased, including the levels of interferon-γ (IFN-γ) and interleukin-17 (IL-17), of which the decrease by OXA treatment was particularly robust. In healthy mice injected with saline instead of LPS, peripherally administered OXA did not affect the cytokine levels in the peripheral blood (*Figure 2*). To examine the inflammatory state of the CNS, we measured the levels of the same set of 32 cytokines in homogenized brains of LPS-injected mice. LPS increased the levels of most cytokines in the brain, and peripherally administered OXA significantly inhibited the increase of certain cytokines in the brain at 4 hr and/or 22 hr after LPS injection, particularly the microglia-derived chemokines such as MIP-1α/1$\beta$ (*Figure 2—figure supplement 2*). These findings suggest that orexin may modify the course of disease in endotoxin shock by suppressing excessive cytokine production.

The excessive cytokine production in sepsis is primarily attributed to macrophages activated by LPS (*Beutler and Rietschel, 2003*; *Moore et al., 1976*). Our findings suggest that orexin may regulate LPS-induced macrophage activation either by acting through the CNS or by acting directly on macrophages. To examine the latter possibility, we prepared peritoneal macrophages from wild-type or OXRKO mice, and then activated cultured macrophages by LPS in the absence or presence of OXA (1 nM and 100 nM). Levels of IL-17, IFN-γ, IL-6, and TNF-α mRNA in cultured macrophages from both wild-type and OXRKO mice were increased after LPS addition. We found no evidence for orexin's direct action on macrophages, however, since the induction of these mRNAs was not affected in the presence of OXA (1 nM and 100 nM) (*Figure 2—figure supplement 3*).

Catecholamines and corticosterone could be the possible immune modulators mediating the anti-inflammatory effects of orexin through the CNS (*Tracey, 2002*). Norepinephrine levels in the serum from LPS-injected mice were significantly decreased, compared to healthy mice (*Figure 1—figure supplement 1A*). OXA increased serum epinephrine and partially inhibited the decrease of norepinephrine levels in LPS-injected mice (*Figure 1D*). Corticosterone levels in the plasma from LPS-injected mice were significantly increased, compared to healthy mice (*Figure 1—figure supplement 1B*). This is consistent with the report that LPS activates PVN (*Elmquist et al., 1996*). OXA further augmented the increase of corticosterone levels in LPS-injected mice (*Figure 1E*). These findings suggest that orexin simultaneously modulates a multitude of peripheral effectors to suppress the systemic inflammatory response.

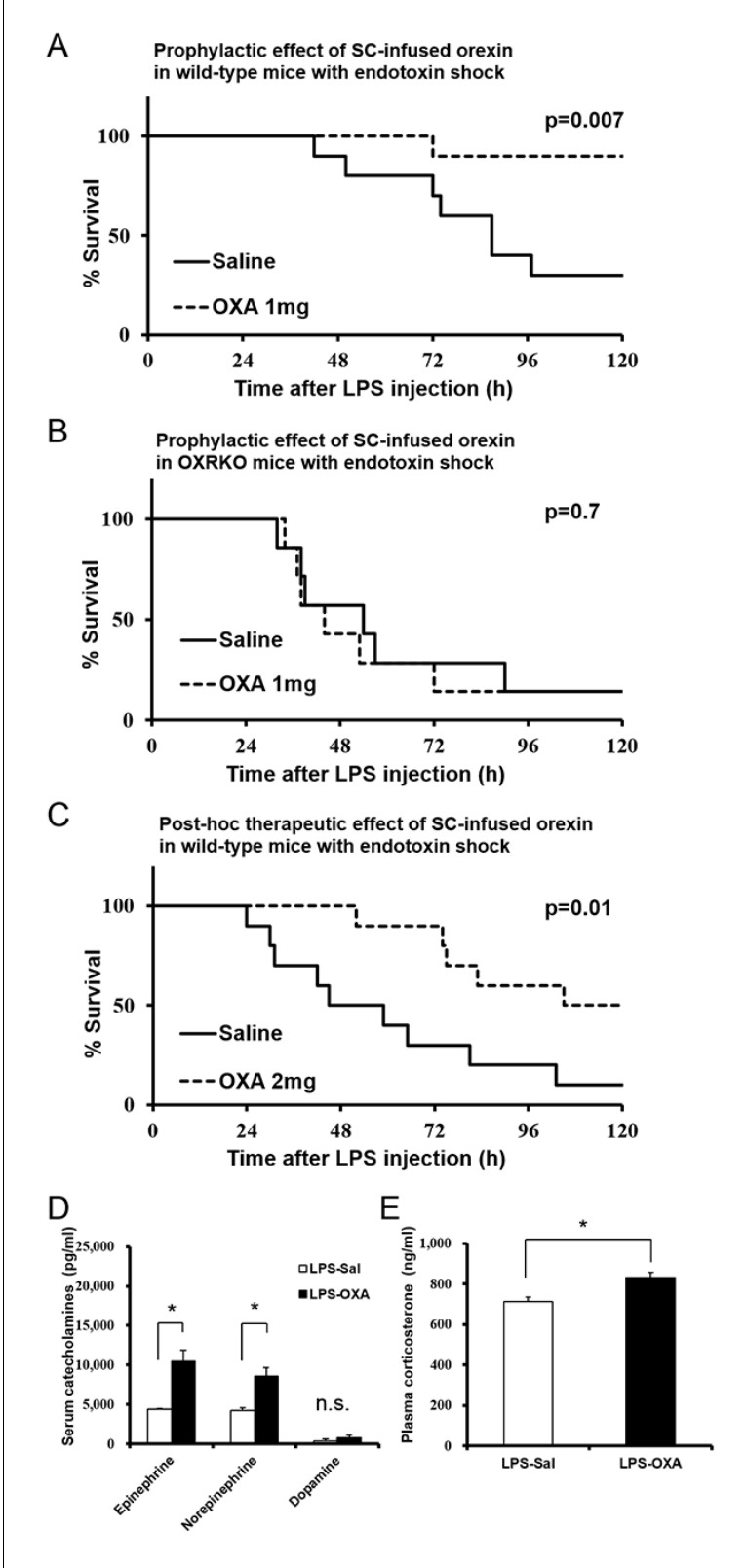

**Figure 1.** Effects of peripherally administered orexin-A (OXA) on survival in mice with endotoxin shock. (**A**) Kaplan-Meier survival curves of wild-type mice subcutaneously (SC) administered with saline or orexin-A (OXA; 1 mg/mouse/24 hr) 30 min before lipopolysaccharide (LPS; 10 mg/kg) injection (each group n = 10). (**B**) Kaplan-Meier survival curves of *Hcrtr1*[−/−];*Hcrtr2*[−/−] (OXRKO) mice SC-administered with saline or OXA (1 mg/mouse/24 hr) 30 min

*Figure 1 continued on next page*

*Figure 1 continued*
before LPS injection (each group n = 7). (**C**) Kaplan-Meier survival curves of wild-type mice SC-administered with saline or OXA (2 mg/mouse/24 hr) 30 min after LPS injection (each group n = 10). (**D, E**) Effect of OXA treatment on the levels of catecholamines (**D**) in the serum and corticosterone (**E**) in the plasma from LPS-injected mice, compared to saline treatment (each group n = 3–5, *p<0.05). Data are presented as mean±s.e.m. Statistical significance assessed by Mantel Cox log-rank test (**A**–**C**) and unpaired t-test (**D, E**). Data are replicated in at least three independent experiments. n.s: not significant.
The following figure supplement is available for figure 1:

**Figure supplement 1.** Serum catecholamines (**A**) and plasma corticosterone (**B**) levels at 22 hr after LPS injection.

## Orexin ameliorates hypothermia and bradycardia, survival markers for mice with endotoxin shock

In rodents (*Blanqué et al., 1996*) and humans (*Kushimoto et al., 2013*), hypothermia is regarded as an important indicator for the severity of inflammation in sepsis. We investigated the correlation between survival rate and body temperature in mice with endotoxin shock. Core body temperature of orexin-administered mice with endotoxin shock remained higher than those of saline-administered mice with endotoxin shock (*Figure 3A*). The mice that survived through endotoxin shock had significantly higher body temperatures at 24 hr post-LPS injection, as compared with non-survivors (*Figure 3B*). A cut-off value of 28.5°C gave an optimal receiver operating characteristic (ROC) curve, predicting survival with highest sensitivity and specificity (*Figure 3C*). This suggests that body temperature at 24 hr after LPS injection can be a good predictor or marker of eventual survival in our mice with endotoxin shock. This does not necessarily mean that maintenance of body temperature is a sole cause of the improved survival in endotoxin shock. Indeed, we found that the survival of mice with endotoxin shock was not improved by simply warming up the animal on a heat pad to the same degree as the orexin-treated mice (*Figure 3—figure supplement 1*), although it is reported that the survival is improved by dramatically elevating body temperature through external heating to 42°C to activate the heat shock protein pathway (*Chu et al., 1997*).

Assessment of cardiovascular function (e.g., heart rate and blood pressure) is also important for the evaluation of the severity of endotoxin shock. We investigated the effects of transiently administered OXA on body temperature and heart rate in mice with endotoxin shock. Body temperature and heart rate of mice ICV-injected with OXA remained significantly higher for 3 hr, compared to saline-injected mice (*Figure 3D,E*). Peripheral (IP) injection of OXA (0.1 mg/mouse) also increased body temperature (*Figure 3F,G*) and heart rate (*Figure 3H,I*) transiently in wild-type mice but not OXRKO mice with endotoxin shock, indicating that this transient effects of orexin are mediated by orexin receptors. In contrast, IP injection of OXA had no effects on body temperature and heart rate in healthy wild-type mice (*Figure 3—figure supplement 2A–D*).

## Peripherally administered orexin penetrates the BBB under endotoxin shock and acts directly on the CNS

Peripherally administered orexin does not efficiently penetrate the BBB under healthy conditions (*Fujiki et al., 2003*). However, we found that the uptake of [$^{125}$I]OXA by the brain after intraperitoneal (IP) administration was significantly increased in mice with endotoxin shock, compared to healthy mice (*Figure 4A,B*). These results indicate that peripherally administered orexin can penetrate the BBB under endotoxin shock, likely due to the dysfunction of BBB induced by systemic inflammation.

We then found that central (ICV) but not peripheral (SC) administration of OXA at a lower dose (0.3 mg/mouse/24 hr) starting 30 min before LPS injection improves the survival (*Figure 4C*), further suggesting a central action of orexin. Here again, although orexin was administered for the first 24 hr only, its life-supporting effects manifested over the next several days. We also investigated the early-phase effects of centrally administered OXA on the levels of relevant cytokines/chemokines (see *Figure 2*). ICV administration of OXA (0.1 mg/mouse/4 hr) significantly suppressed the increase in the level of MIP-1α and showed a tendency to suppress the increase in the levels of MIP-1β and TNF-α in the peripheral blood at 4 hr after LPS injection (*Figure 4D*). ICV administered OXA simultaneously

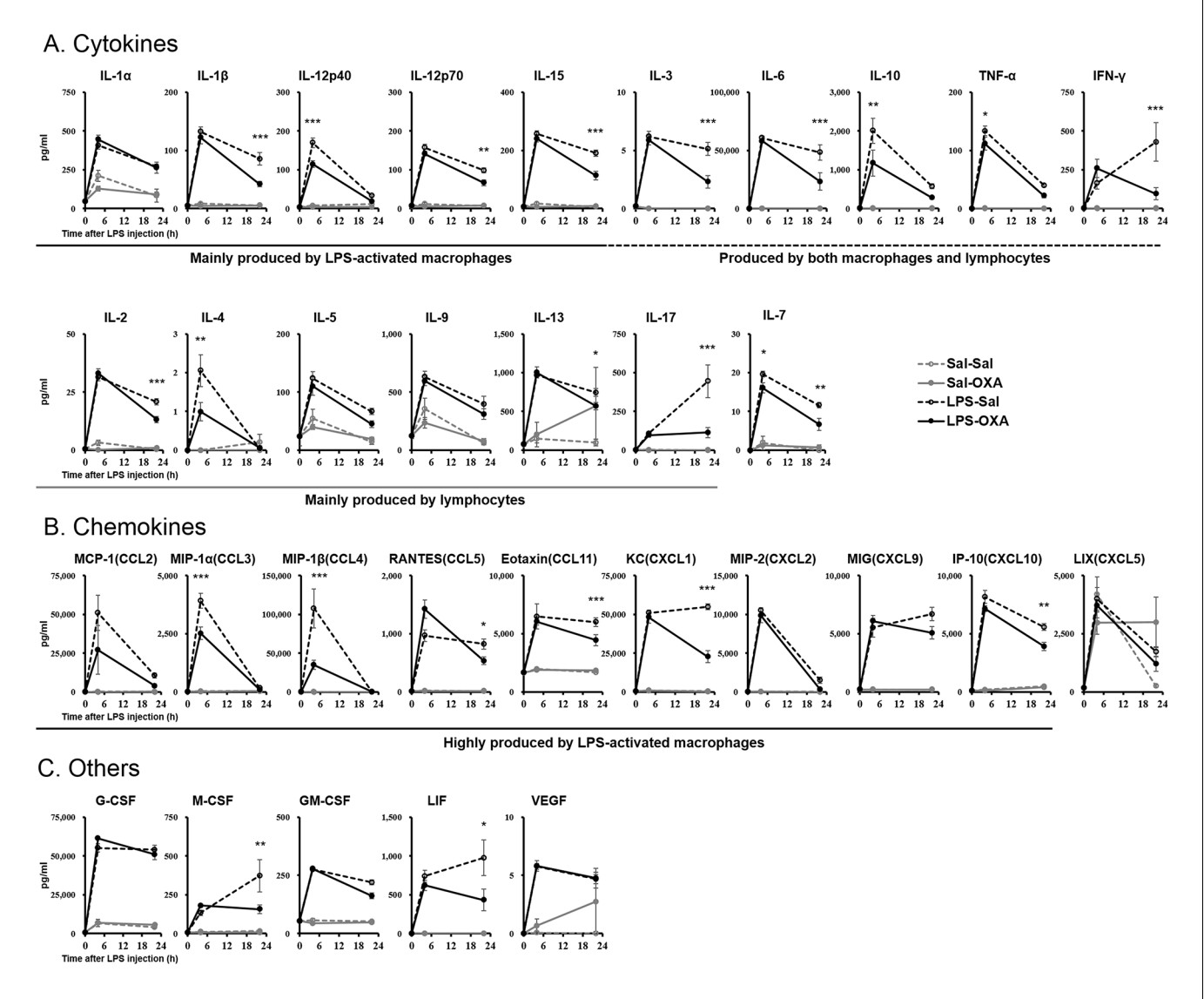

**Figure 2.** Effects of peripherally administered OXA on cytokine production in mice with endotoxin shock. Effects of OXA treatment on the levels of 32 cytokines in serum at 4 hr and 22 hr after injection of LPS (LPS-OXA) or saline (Sal-OXA), compared to saline treatment (LPS-Sal, Sal-Sal) (each group n = 8, *p<0.05, **p<0.01, ***p<0.001). OXA (1 mg/mouse/24 hr) started to be SC-administered at 30 min before LPS injection. Statistical significance assessed by 2-way ANOVA coupled with the Bonferroni's test. Data are replicated in at least three independent experiments.

The following figure supplements are available for figure 2:

**Figure supplement 1.** Effects of SC-infused OXA on body temperature at 4 hr or at 22 hr after LPS injection.

**Figure supplement 2.** Effects of SC-infused OXA on cytokine production in brain at 4 hr or at 22 hr after LPS injection.

**Figure supplement 3.** Effects of OXA on expressions of IL-6 (A), TNF-α (B), IL-17 (C), and IFN-γ (D) mRNA in cultured peritoneal macrophages from wild-type and OXRKO mice.

ameliorated hypothermia induced by LPS (*Figure 4—figure supplement 1*). These findings suggest that orexin modifies the course of inflammatory processes in endotoxin shock by acting directly on the CNS.

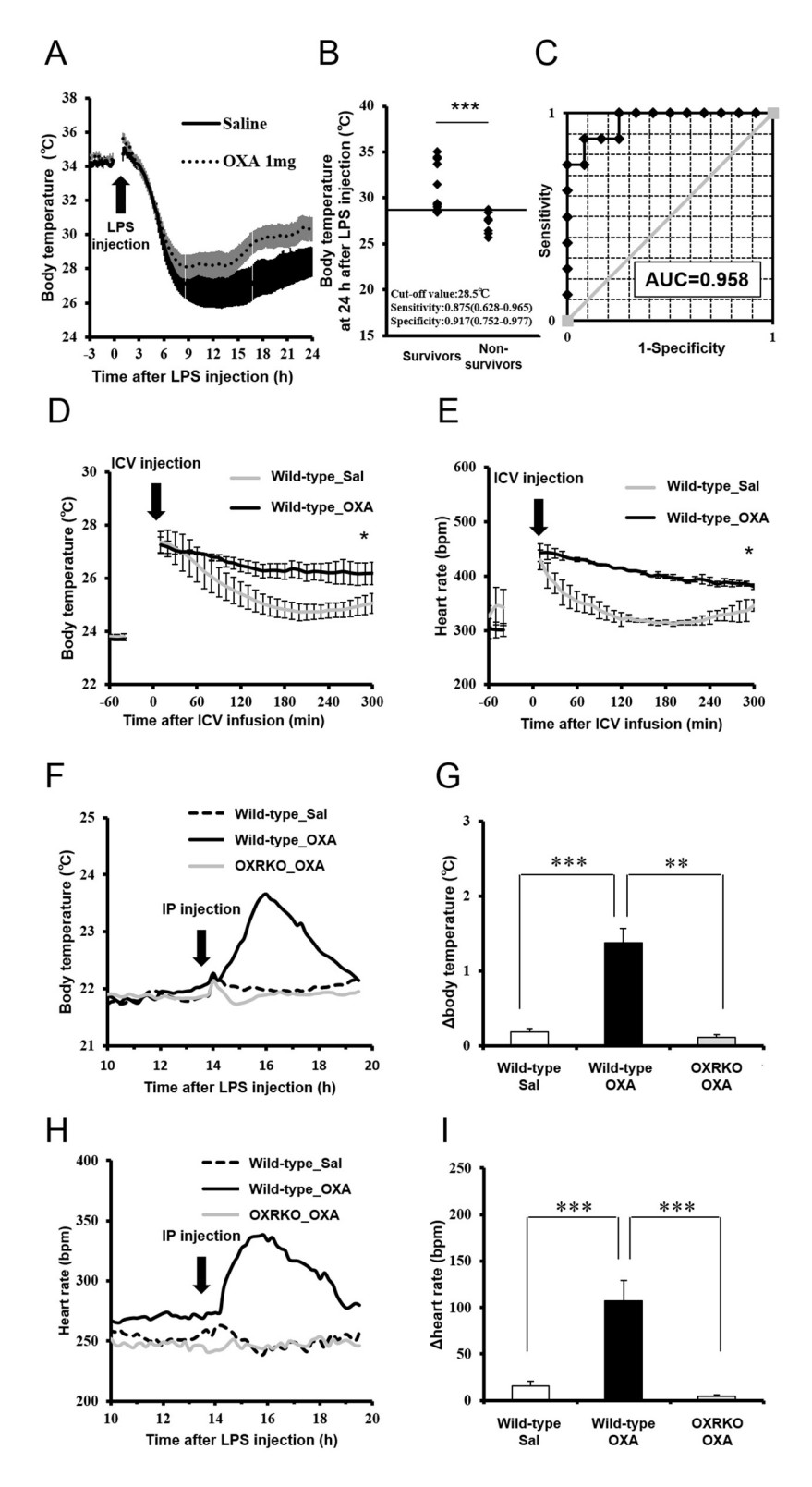

**Figure 3.** Effects of OXA on body temperature and heart rate in mice with endotoxin shock. (**A**) The changes in body temperature of LPS-injected mice treated with OXA (1 mg/mouse/24 hr) or saline (each group n = 10). (**B**) The correlation between the survival and body temperature in mice with endotoxin shock (Survivor n = 12, Non-survivor n = 8, ***p<0.001). (**C**) Receiver operating characteristic (ROC) curve between survival and body temperature in mice with endotoxin shock. AUC: area under the curve. (**D, E**) Transient effects of bolus ICV-injected OXA (Wild-type_OXA) or saline

*Figure 3 continued on next page*

*Figure 3 continued*

(Wild-type_Sal) on body temperature (D) and heart rate (E) in LPS-injected wild-type mice (each group n = 4, *p<0.05). (F, H) Transient effects of bolus IP-injected OXA on body temperature (F) and heart rate (H) in mice with endotoxin shock. (G, I) IP injection of OXA (Wild-type_OXA) but not saline (Wild-type_Sal) increased body temperature (G) and heart rate (I) transiently in LPS-injected wild-type mice, but not in LPS-injected OXRKO mice (OXRKO_OXA) (each group n = 4, **p<0.01, ***p<0.001). Statistical significance assessed by unpaired t-test (B), 2-way ANOVA (D, E), and 1-way ANOVA coupled with the Bonferroni's test (G, I). Data are replicated in at least three independent experiments.

The following figure supplements are available for figure 3:

**Figure supplement 1.** Effects of whole-body warming with a heat pad on body temperature (upper) and survival rate (lower) in mice with endotoxin shock.

**Figure supplement 2.** Transient effects of IP-injected OXA on body temperature and heart rate in healthy wild-type mice injected with saline (A, B, C, D) or in *Hcrtr1*[-/-] (OX1RKO) and *Hcrtr2*[-/-] (OX2RKO) mice with endotoxin shock (E, F, G, H).

## Orexin's targets in the CNS

To clarify the central targets of orexin's life-supporting effects in mice with endotoxin shock, we then focused on the transient thermogenic effect of orexin as a surrogate index for the survival. Raphe nuclei, containing serotonergic neurons, exist in the pons/medulla and midbrain. Medullary raphe nuclei are known as one of the important sites of action for orexin's thermogenic effect (*Tupone et al., 2011*). In wild-type mice with endotoxin shock, IP administration of OXA (0.1 mg/mouse) activated serotonergic neurons in the medullary raphe pallidus and raphe magnus (RPa/RMg), as assessed by double immunostaining with anti-Fos and anti-5HT antibodies (*Figure 5A*). There was a significant increase in the percentage of Fos-positive serotonergic neurons in RPa/RMg after OXA administration, compared to saline administration (*Figure 5B*). This Fos induction was not seen in healthy wild-type mice (*Figure 5B*) or in OXRKO mice with endotoxin shock (*Figure 5C*). Midbrain dorsal raphe nucleus (the dorsal raphe nucleus, dorsal part and ventrolateral part; DRD/DRVL) is reported to be a thermosensor activated by elevation of body temperature (*Hale et al., 2011*). In wild-type mice with endotoxin shock, there was also a significant increase of Fos activity in serotonergic neurons in DRD/DRVL by OXA administration (*Figure 5—figure supplement 1A–C*). We systematically surveyed other known orexin targets in the CNS, and detected no statistically significant changes in Fos activities upon OXA administration under endotoxin shock in these areas (*Figure 5—figure supplement 1D–H*). Neurons in locus coeruleus (LC), rostral ventrolateral medulla (RVLM, a sympathetic nucleus), nucleus tractus solitaries (NTS, a parasympathetic sensory nucleus), and dorsal motor nucleus of the vagus (DMNX, a parasympathetic motor nucleus) are known to be activated after LPS injection (*Elmquist et al., 1996*). Stimulation of LC by corticotropin-releasing hormone is reported to exert an immunosuppressive effect on the peripheral lymphocytes (*Rassnick et al., 1994*). We confirmed that these neurons were activated after LPS injection (*Figure 5—figure supplement 2A*), independently of orexin signaling (*Figure 5—figure supplement 2B*). We concluded that, in mice with endotoxin shock, peripheral administration of orexin activated the serotonergic thermoregulatory system in an orexin receptor-dependent manner.

## Discussion

Septic shock is a life-threatening condition with no effective treatment reported so far (*Peake et al., 2014*; *Mouncey et al., 2015*; *Yealy et al., 2014*; *Ranieri et al., 2012*). We found that the survival of mice with endotoxin shock is improved by central administration of orexin. Moreover, surprisingly, peripheral injection of orexin also improved survival, because orexin could penetrate into the CNS due to BBB dysregulation under the condition of systemic inflammation. Our findings show that orexin could be a new therapeutic agent for septic shock; we propose here an innovative approach to target the CNS for systemic inflammation, employing an unexpected method to deliver neuropeptides into the brain by taking advantage of the pathophysiology of sepsis.

In mice with endotoxin shock, peripherally administered orexin robustly decreased the levels of IL-17 and IFN-γ in peripheral blood along with other pro-inflammatory cytokines such as IL-6 and TNF-α. Inhibiting either IL-17 or IFN-γ by neutralizing antibodies within 24 hr after cecal ligation and

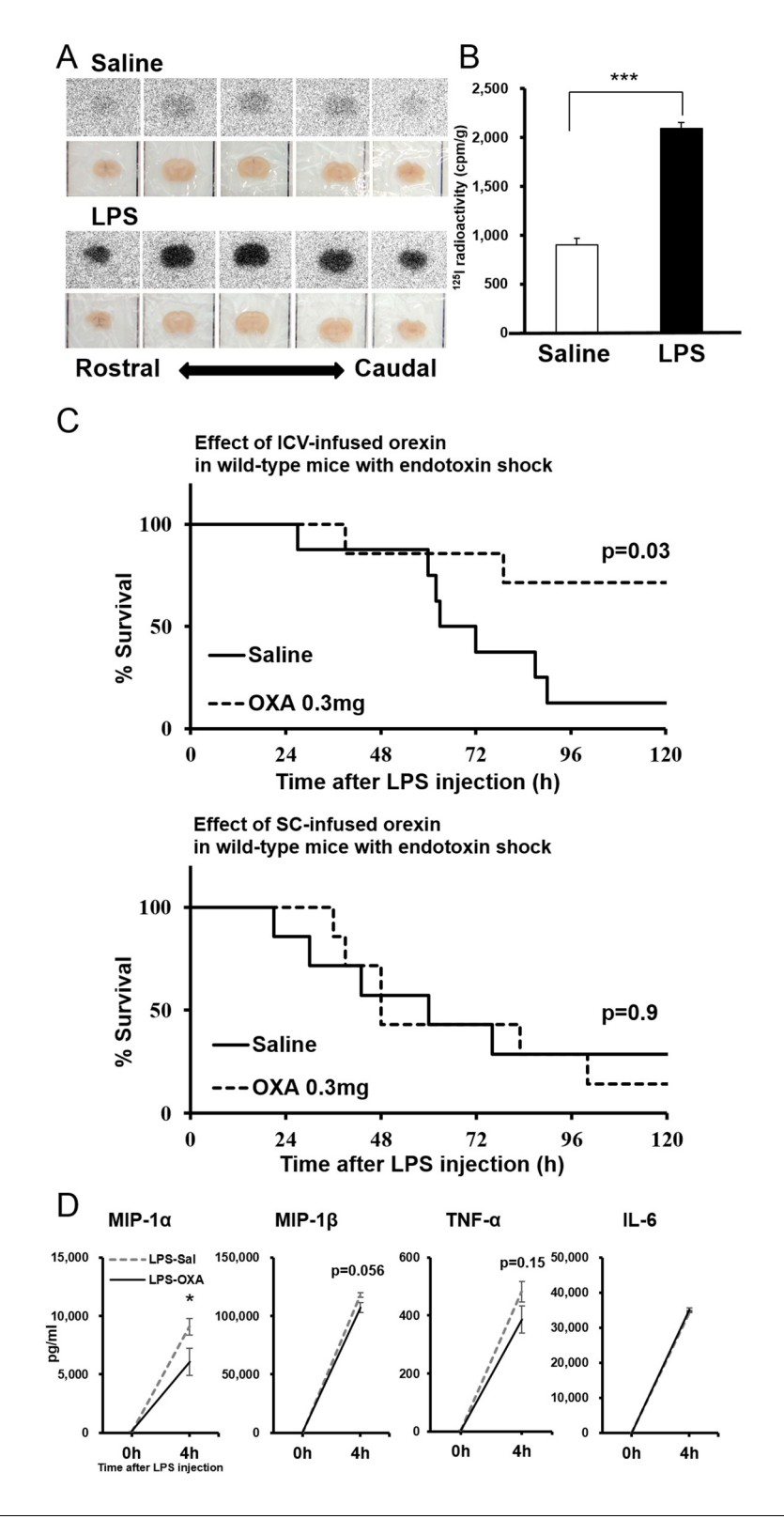

**Figure 4.** Direct action of OXA on the CNS in mice with endotoxin shock. (**A**) [$^{125}$I]OXA-autoradiography of 1 mm coronal brain sections from control (saline) and endotoxin shock (LPS) mice. The brains were removed without perfusion at 2 hr after intraperitoneal (IP) administration of [$^{125}$I]OXA, and were fixed in 4% PFA overnight. The sections were exposed to imaging plates for five days, and then scanned by BAS-2500 (Fuji Film). (**B**) The levels of radioactivity in the whole brain from LPS- or saline-injected mice at 2 hr after IP administration of [$^{125}$I]OXA (each group n = 4, ***p<0.001). (**C**) Kaplan-
*Figure 4 continued on next page*

*Figure 4 continued*

Meier survival curves of wild-type mice intracerebroventricularly (ICV, upper; saline: n = 8, OXA: n = 7) or subcutaneously (SC, lower; each group n = 7) administered with saline or OXA (0.3 mg/mouse/24 hr) before LPS injection. (D) Effects of ICV-administered OXA on the levels of MIP-1α, MIP-1β, TNF-α and IL-6 in serum at 4 hr after LPS injection (LPS-OXA), compared to saline treatment (LPS-Sal) (each group n = 8, *p<0.05). OXA administration (0.1 mg/mouse/4 hr) started 30 min before LPS injection. Data are presented as mean±s.e.m. Statistical significance assessed by unpaired t-test (B), Mantel Cox log-rank test (C), and 2-way ANOVA coupled with the Bonferroni's test (D). Data are replicated in at least three independent experiments.

The following figure supplement is available for figure 4:

**Figure supplement 1.** Effects of ICV-infused OXA on body temperature at 4 hr after LPS injection.

puncture (CLP) is reported to improve survival (*Flierl et al., 2008*; *Márquez-Velasco et al., 2011*). Although these antibodies are administered in the early period only, their life-supporting effects manifest over the next several days, similar to the effects of orexin in the present study. IL-17 produced during sepsis is implicated in the 'cytokine storm,' in which the inflammatory response is amplified, increasing the lethality of septic shock (*Flierl et al., 2008*). IL-17 synergistically potentiates LPS-induced activation of peritoneal macrophages in vitro and increases the levels of IL-6, TNF-α, and IL-1β. In our study, orexin did not directly inhibit the increase of cytokine production in LPS-activated peritoneal macrophages, although TNF-α production in LPS-stimulated microglia, the primary immune cells in the CNS, is reported to be decreased by OXA pretreatment (*Xiong et al., 2013*). Together with the findings that centrally administered orexin was effective at a lower dose in improving the survival of mice with endotoxin shock and in suppressing MIP-1α levels in their peripheral blood, our results indicate that the anti-inflammatory effects of orexin are through its action in the CNS rather than in the peripheral tissues.

We found that orexin increases blood corticosterone levels of mice with endotoxin shock, which is consistent with the report that ICV administration of orexin increases the expression of ACTH in CLP-induced septic shock model (*Deutschman et al., 2013*). The blood corticosterone level is increased in mice with LPS-induced lung injury (*Landgraf et al., 2014*), and even a slight upregulation of corticosterone by exogenous leptin may improve LPS-induced lung injury (*Landgraf et al., 2014*). Therefore, the increase of corticosterone levels in endotoxin shock may play a role in the anti-inflammatory effects of orexin, although further studies with glucocorticoid agonists/antagonists are necessary to substantiate this possibility. Indeed, glucocorticoids are sometimes used for human patients with sepsis and effective in some cases (*Dellinger et al., 2013*).

We also found that orexin increased the levels of epinephrine and norepinephrine in the blood of mice with endotoxin shock: Further studies with adrenergic antagonists would fully qualify the role of these neuroendocrine changes in the observed improvements in survival. The increases of catecholamines might not only stabilize hemodynamics by potentiating cardiovascular function, but also modulate inflammation through adrenoceptors (*Spengler et al., 1990*; *Vida et al., 2011*). However, although catecholamines are used to stabilize hemodynamics in human patients with sepsis (*Dellinger et al., 2013*), their effects are often limited because vascular responsiveness to catecholamines is reduced during sepsis (*Parker and Parrillo, 1983*). Also, adrenergic modulation of inflammation is complicated. The adrenergic anti-inflammatory effects are mostly mediated by β2-adrenoceptors (*Vida et al., 2011*), while stimulation of α2-adrenoceptors induces pro-inflammatory effects (*Spengler et al., 1990*). Moreover, systemically administered catecholamines act on all tissues without coordination. In contrast, catecholamines endogenously released under the CNS command are localized in the specific target tissues and their actions on each target are integrated by the CNS. This may be why the life-supporting effects of orexin administration on sepsis may be longer lasting and more effective than those of catecholamine administration alone. Indeed, in the rat septic shock model, administration of norepinephrine for 24 hr improves the survival rate by only 20% (*Li et al., 2009*). In contrast, administration of orexin for 24 hr showed 70–80% and 40% improvements in the survival rate for prophylactic and post-hoc protocols, respectively, in our study (*Figures 1* and *3*). Furthermore, it is reported that the treatment with both catecholamines and glucocorticoids synergistically attenuates the production of pro-inflammatory cytokines and IL-17 during sepsis (*Bosmann et al., 2013*).

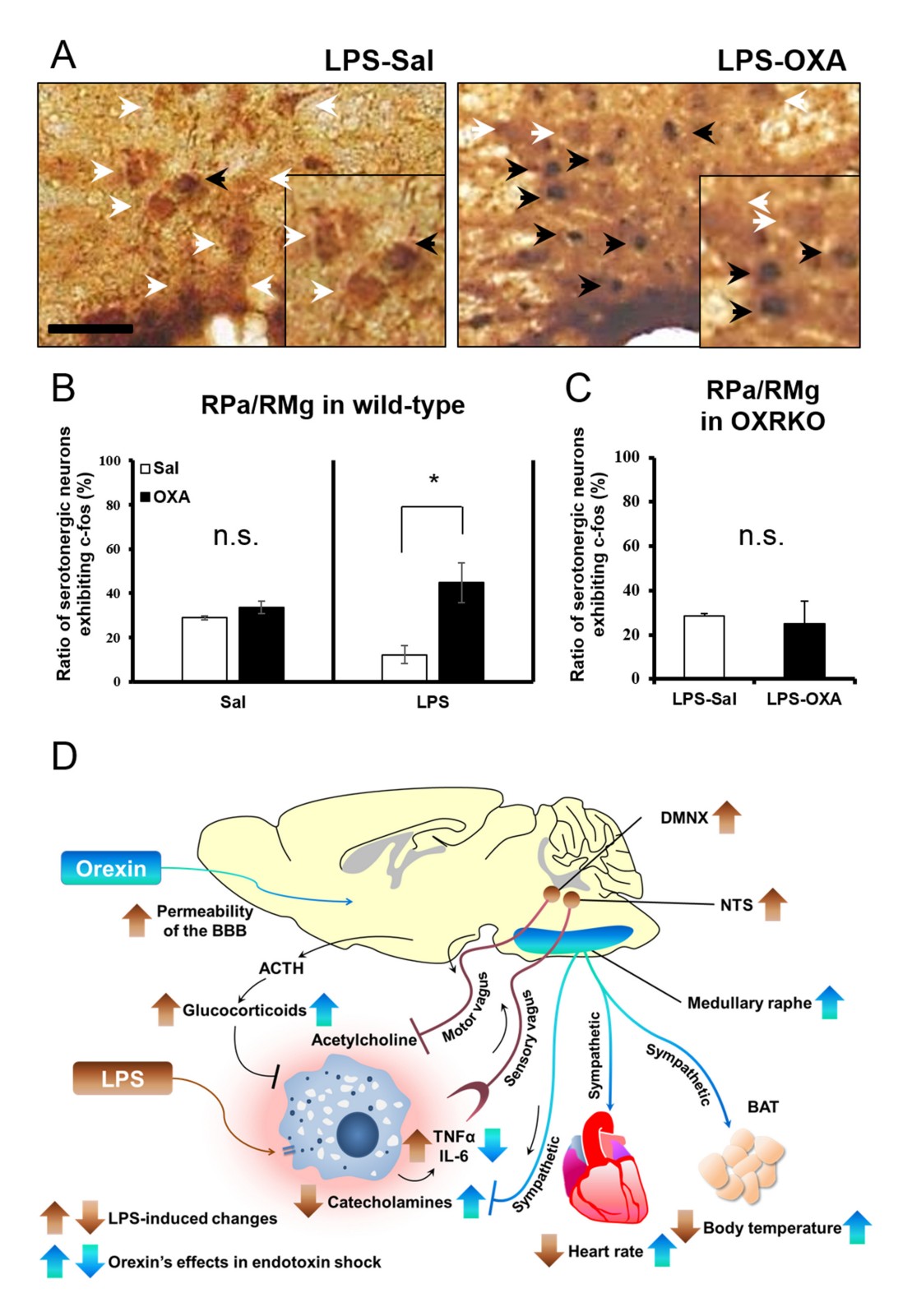

**Figure 5.** The hypothetical schema of multiple pathways by which OXA may improve survival in mice with LPS-induced endotoxin shock (see also main text). (**A**) IP-administered OXA (LPS-OXA, right) but not saline (LPS-Sal, left) activated serotonergic neurons in the raphe pallidus nucleus and raphe magnus nucleus (RPa/RMg) in LPS-injected wild-type mice by immunohistochemistry using anti-c-fos and anti-5HT antibodies. Black arrows indicate double positive cells and white arrows indicate 5HT-single positive cells. (**B, C**) IP-administration of OXA significantly activated serotonergic neurons in

*Figure 5 continued on next page*

*Figure 5 continued*

RPa/RMg of LPS-injected wild-type mice, compared to saline administration (each group n = 4, *p<0.05) (B). There was no significant difference in the activities of serotonergic neurons in RPa/RMg between saline and OXA administration in healthy wild-type mice (B) or in OXRKO mice with endotoxin shock (C). Statistical significance assessed by unpaired t-test. Scale bar: 100 μm. Data are replicated in at least three independent experiments. (D) LPS activates macrophages via toll-like receptor 4 and produces TNF-α, IL-6, and other cytokines. These cytokines activate NTS and DMNX through sensory vagus nerves (*Tracey, 2002*). Activated motor vagus nerves regulate the inflammatory response and decrease heart rate and body temperature. Orexin sustains heart rate and body temperature through sympathetic nervous system by activating medullary raphe, and also regulates the inflammatory response by central actions. Orexin thus improves the survival through a multitude of pathways, including neuroendocrine and autonomic nervous systems. NTS: nucleus tractus solitaries. DMNX: dorsal motor nucleus of the vagus.

The following figure supplements are available for figure 5:

**Figure supplement 1.** Systematic survey of orexin's targets in the CNS of mice with endotoxin shock.

**Figure supplement 2.** Activation of neurons in the LC, RVLM, NTS and DMNX at 3 hr after LPS injection in wild-type (A) or OXRKO (B) mice.

There has been no report showing the role of the raphe nuclei in septic shock. Raphe nuclei are reported to be an orexin-regulated region for brown adipose tissue thermogenesis and heart rate modulation (*Tupone et al., 2011*). Taken together, our findings show that orexin improves the survival of mice with endotoxin shock through a multitude of pathways, including the neuroendocrine and autonomic nervous system, by targeting the CNS (hypothesized in *Figure 5D*). The precise mechanism through which CNS orexin modulates inflammatory responses merits further investigation. The present study suggests that a coordinated intervention through the CNS against the complex pathophysiology of systemic inflammation can be a novel therapeutic approach for septic shock. While we therapeutically exploited the leakiness of BBB under sepsis in this study, BBB dysfunction may also permit cytokines and inflammatory cells to freely enter into the CNS, which may constitute an additional element of pathophysiology of sepsis, such as 'sepsis-associated encephalopathy' (*Widmann and Heneka, 2014*). The failure of traditional therapy (e.g., catecholamines) in septic shock has been commonly attributed to defects in peripheral response mechanisms. Our study suggests that it may be the CNS regulatory machinery that is dysfunctional, which could be partially restored by orexin.

## Materials and methods

### Animals

Eight- to 10-week-old male mice from four genotypes ($Hcrtr1^{-/-};Hcrtr2^{-/-}$, $Hcrtr1^{-/-}$, $Hcrtr2^{-/-}$, and wild-type, on a C57BL/6J background, generated by crosses between homozygous mice) were randomly assigned to experimental groups. Experimental mice were individually housed and kept on a 12 hr:12 hr light:dark schedule at an ambient temperature of 23 ± 1°C under specific pathogen-free conditions.

### Solutions and materials

Lipopolysaccharide (LPS, Lipopolysaccharides from Escherichia coli 055:B5, Sigma, St. Louis, MO) was dissolved in saline (Otsuka, Japan) and LPS solution was adjusted to 1 mg/ml. Human orexin-A (Peptide Institute, Japan) was dissolved in saline and adjusted for each experiment. Phosphate Buffered Saline (PBS) was made from Phosphate Buffered Saline Powder (0.01 mol/l, pH 7.2–7.4, Wako, Japan) and Phosphate Buffer (PB) was made from Phosphate Buffered Powder (1/15 mol/l, pH 7.4, Wako). Paraformaldehyde (PFA, Nacalai, Japan) was dissolved in PBS and adjusted to 4% (w/w).

### LPS-induced model of endotoxin shock in mice

To induce the endotoxin shock, mice were intraperitoneally (IP) injected with LPS (10 mg/kg) at ZT10.5.

## Surgery

To investigate the central effect of orexin, mice were implanted with a guide cannula into the left lateral ventricle. To monitor body temperature and heart rate of mice continuously every ten minutes, the mice were subcutaneously implanted with a telemetry probe (TA11TA-F10, Data Science International, St. Paul, MN). Mice were recovered for one week after operation.

## Survival experiments in endotoxin shock

To investigate the prophylactic effect of peripherally administered orexin on survival in mice with endotoxin shock, mice were implanted subcutaneously with an osmotic pump (Alzet, Durect, Cupertino, CA) containing 1 mg orexin-A dissolved in 200 µl saline (or containing 200 µl saline as control) under anesthesia before LPS injection. Subcutaneous infusion started at a rate of 8 µl/hr for 24 hr at the same time of implantation of the pump. To investigate the post-hoc therapeutic effect of peripherally administered orexin on survival, mice were implanted subcutaneously with a programmable microinfusion pump (iPRECIO, SMP300, Primetech, Japan) containing 1 mg orexin-A dissolved in 100 µl saline (or containing 100 µl saline as a control) under anesthesia at 30 min before LPS injection. These pumps are wirelessly controlled by an external scheduler. The infusion schedule was programmed with iPRECIO Management Software ver 1.1. At 30 min after LPS injection, subcutaneous infusion started at a rate of 8 µl/hr for 24 hr. The next day iPRECIO was refilled with orexin A (1 mg orexin-A dissolved in 100 µl saline) or 100 µl saline. To investigate the effect of centrally administered orexin on survival, mice were intracerebroventriculerly (ICV) administered with 0.06 mg orexin-A dissolved in 3 µl saline (or 3 µl saline as a control) in 5 min and implanted subcutaneously with an osmotic pump containing 200 µl saline as transfusion (at a rate of 8 µl/hr) under anesthesia at 30 min before LPS injection. At the same time of LPS injection, ICV infusion was started at the rate of 0.5 µl /hr (orexin-A concentration was 0.02 mg/µl). To investigate the effect of warming up to the same degree as orexin-treated mice on survival, mice were warmed on a heat pad at 30°C for 24 hr at the same time of LPS injection. In all the survival experiments, body temperature was monitored for seven days after LPS injection. We assessed survival as primary endpoint, and body temperature at 24 hr after LPS injection as secondary endpoint. Observation was stopped in seven days or by death. Relationship between survival and body temperature was expressed by the receiver operating characteristic (ROC) curve, plotting true positive rate (Y-axis; sensitivity) against the false positive rate (X-axis; one minus specificity) at various body temperature values. The ROC curve was assessed by area under the curve (AUC), and an optimal cut-off values was determined, which gave maximum AUC. The sample size was calculated on the following assumptions; α is 0.05, power is 0.8, the usual survival rate in control group at two days is 0.2 and the expected survival rate in the treatment group at seven days is 0.6–0.9. All the experimental mice were used as data samples without inclusion/exclusion.

## Examination of BBB permeability

Monitoring body temperature, LPS- or saline-injected mice were IP-administered with [125I]orexin-A at 10 (Borovikova et al., 2000) cpm. At 2 hr after [125I]orexin-A administration, we removed the whole brain and the levels of radioactivity were determined in a γ-counter. The brains were then fixed in 4% PFA overnight. One-mm coronal sections were exposed to imaging plates (Bas-SR 2040, Fuji Film, Japan) for five days and then scanned by BAS-2500 (Fuji Film).

## Measurement of cytokines/chemokines, catecholamines, and corticosterones

### Sampling of blood and tissues in endotoxin shock

To investigate the effects of peripherally or centrally administered orexin on cytokine production in mice with endotoxin shock, orexin was administered as described above. Monitoring body temperature, the blood and brain were sampled at 4 hr or 22 hr after LPS injection. For sampling the blood and brain, mice were anesthetized with isoflurane. Blood was sampled from the left ventricle using a 27 G needle (Terumo, Japan) with heparin or EDTA-2Na, and was centrifuged at 3000 rpm for 15 min. The supernatants were sucked-up as blood plasma or serum. Removed brains were frozen in liquid nitrogen and were stored at −80°C.

## Protein extraction from brain tissues

A single whole brain was weighed and homogenized with Tissue Rupture (Qiagen, Germany) in a 5x volume of extraction buffer (20 mM TrisHCl, 0.15 M NaCl, 0.05% Tween-20 and Protease Inhibitor Cocktail; Sigma). The samples were centrifuged (1000 g) for 10 min at 4°C. The supernatant was removed and centrifuged a second time (20,000 g for 40 min at 4°C). Protein levels of all samples were quantified with a BCA Protein Assay Kit (Thermo Scientific, Waltham, MA).

## Measurement of cytokines/chemokines

Cytokines/chemokines were measured by the Multiplex system (Millipore, Germany). Plasma samples were diluted 1:2 in the assay buffer provided in the kit. For measuring cytokines in brain tissue, 250 μg of protein was used. Twenty-five μl of sample and standard were incubated with microspheres containing antibodies in each well overnight at 2–8°C. The analytes on the surface of the microspheres were then detected by a cocktail of biotinylated antibodies. Following binding of streptavidin-phycoerythrin conjugate, the reporter fluorescent signal was measured with a Luminex200 reader (Millipore).

## Measurement of catecholamines and corticosterones

Measurement of serum catecholamines and plasma corticosterone were performed by SRL. Inc.

## Measurement of cytokine mRNA levels in cultured peritoneal macrophages

Mouse peritoneal macrophages were harvested five days after IP injection of 1.5 ml thioglycolate (2.4 g/100 ml; BD Biosciences, Franklin Lakes, NJ) by peritoneal lavage with PBS and the purity of cell suspension was >95% macrophages. Macrophages ($5 \times 10^{-6}$ per experimental condition) were allowed to adhere to tissue culture plates overnight. Cells were incubated in 10% FBS/DMEM containing LPS (50 ng/ml) in the absence or presence of orexin A ($10^{-9}$, $10^{-7}$ M) for 6 hr at 37°C. Then culture medium was sampled and total RNA was prepared by using RNeasy (Qiagen). The expression levels of IL-6, IL-17, IFN-γ, and TNF-α were measured by QPCR using the following primers:

5′-CACAGAGGATACCACTCCCAACA-3′ and 5′-TCCACGATTTCCCAGAGAACA-3′ for IL-6; 5′-GAAGGCCCTCAGACTACCTCAA 3′ and 5′-TCATGTGGTGGTCCAGCTTTC-3′ for IL-17; 5′-CGCC TATCTTCGGGATGAATC 3′ and 5′-CCAACCGATACTCCATGAAAATG-3′ for IFN-γ; 5′-GGCCTCCC TCTCATCAGTTC-3′ and 5′-GACAAGGTACAACCCATCGGC-3′ for TNF-α.

## Transient response to orexin in endotoxin shock

To investigate transient effects of peripherally administered orexin, mice were bolus IP-administered with 0.1 mg (30 nmol) orexin-A dissolved in 100 μl saline (or with 100 μl saline as a control) at 13.5 hr after LPS or saline injection. To investigate transient effects of centrally administered orexin, mice were ICV-administered with 0.01 mg (3 nmol) orexin-A dissolved in 6 μl saline (or with 6 μl saline as a control) in 6 min at 13.5 hr after LPS injection. Body temperature and heart rate were continuously measured for 6 hr after orexin treatment.

## Immunohistochemistry

Wild-type and OXRKO mice were bolus IP-injected with 0.1 mg (30 nmol) orexin-A dissolved in 100 μl saline (or with 100 μl saline as a control) at 13.5 hr after LPS or saline injection. At 1.5 hr after orexin treatment, mice were anesthetized with pentobarbital sodium (50 mg/kg, IP) and were perfused with PBS followed by 4% PFA. Brains were removed from the skull and were post fixed with 4% PFA overnight and then immersed in 30% sucrose dissolved in PB until they had sunk. After freezing of the brains in Optical Cutting Temperature Compound (O.C.T. Compound, Sakura Finetek, Torrance, CA), the brains were cut serially at 40 μm by the cryostat (Leica, Germany). All sections were incubated with rabbit anti-Fos antibody (1:10,000; RRID: AB_2314421) overnight at 4°C. After incubating with primary antibodies, sections were incubated in biotinylated anti-rabbit IgG antibody (1:200; RRID: AB_2313606) for 1.5 hr and were treated with avidin–biotin complex (ABC kit, Vector Labs., Burlingame, CA) for 30 min at room temperature. The immunoreactive product was visualized in a solution of 3,3′-diaminobenzidine (DAB, Vector Labs.) with nickel and $H_2O_2$, to label the nuclei of the Fos-positive cells in dark black. For identifying the neurochemical feature of

the neurons, after staining for Fos, the sections in the raphe nuclei were incubated with goat anti-5HT antibody (1:5,000; RRID: AB_572262), the ones in the lateral hypothalamus (LH) were incubated with goat anti-orexin-A antibody (1:50; RRID: AB_653610), and the ones in the tuberomammillary nucleus (TMN) were incubated with rabbit anti-histidine decarboxylase (HDC) antibody (1:1,000; RRID: AB_1002154) overnight at 4℃. After incubating with primary antibodies, sections were incubated secondly with each biotinylated antibody (anti-goat IgG antibody (1:200; RRID: AB_2336123) and anti-rabbit IgG antibody (1:200; RRID: AB_2313606)) for 1.5 hr, and ABC solution for 30 min at room temperature. The immunoreactive product was visualized in a solution of DAB without nickel intensification to label the cytoplasm of the cells with the neurochemical feature in light brown. Finally, sections were mounted on slides, air dried, dehydrated and cover-slipped with mounting medium for microscope preparation. Cell counts were conducted by three investigators blind to the assignment of the treatment group by using an LSM700 microscope (Zeiss, Germany). Cytoplasm-labeled cells and double-labeled cells were counted in the sections of each region. We assessed the ratio of double-labeled cells to cytoplasm-labeled cells plus double-labeled cells.

## Statistical analysis

Survival rates were expressed by a Kaplan-Meier curve and comparisons of survival curves were performed with a Mantel Cox log-rank test using PRISM Ver.5.0 (RRID: SCR_002798). All values are expressed as means ± s.e.m. To compare two groups, data were analyzed with unpaired student t-test. To compare more than two groups, data were analyzed with a 1-way ANOVA, and individual group means were then compared with a Bonferroni's test. To compare the time-elapsed data, the data were analyzed with a 2-way ANOVA, and individual group means were then compared with a Bonferroni's test. Differences were considered significant when $p < 0.05$.

## Acknowledgements

We thank Dr. Kaspar Vogt and Dr. Tito Akindele for reading the manuscript. We thank Ms. Yukiko Ishikawa for maintaining the animals. This work was supported by World Premier International Research Center Initiative (WPI), MEXT, Japan, JSPS KAKENHI Grant Number 26220207, and the Funding Program for World-Leading Innovative R and D on Science and Technology (FIRST Program). MY is a former Investigator of Howard Hughes Medical Institute.

## Additional information

### Funding

| Funder | Grant reference number | Author |
|---|---|---|
| Ministry of Education, Culture, Sports, Science, and Technology | WPI | Masashi Yanagisawa |
| Japan Society for the Promotion of Science | KAKENHI 26220207 | Masashi Yanagisawa |
| Japan Society for the Promotion of Science | FIRST Program | Masashi Yanagisawa |

The funders had no role in study design, data collection and interpretation, or the decision to submit the work for publication.

### Author contributions

YO, Conceptualization, Data curation, Formal analysis, Validation, Investigation, Visualization, Methodology, Writing—original draft, Project administration, Writing—review and editing; YI-T, Conceptualization, Data curation, Formal analysis, Supervision, Validation, Investigation, Visualization, Methodology, Writing—original draft, Project administration, Writing—review and editing; NM, Data curation, Formal analysis, Supervision, Validation, Investigation, Visualization, Methodology; MKi, YI, NH, HT, SU, SK, MKa, MM, Data curation, Validation, Investigation; MH, Data curation, Formal analysis, Supervision, Validation, Investigation, Methodology; ST, Supervision, Investigation, Project administration; YH, Formal analysis, Supervision, Investigation, Project administration; MY,

Conceptualization, Resources, Supervision, Funding cquisition, Writing—original draf, Project administration, Writing—review and editing

## Author ORCIDs
Masashi Yanagisawa, http://orcid.org/0000-0002-7358-4022

## Ethics
Animal experimentation: Animal experiments were carried out in a humane manner after receiving approval from the Institutional Animal Care and Use Committee of the University of Tsukuba, and in accordance with the Regulation for Animal Experiments in our university and Fundamental Guideline for Proper Conduct of Animal Experiments and Related Activities in Academic Research Institutions under the jurisdiction of the Ministry of Education, Culture, Sports, Science and Technology (MEXT). Permit Nmber: 16-081

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
