## [Decision Letter]

Thank you for submitting your article "Peripherally administered orexin improves survival of mice with endotoxin shock by acting in the central nervous system" for consideration by *eLife*. Your article has been favorably evaluated by Gary Westbrook as the Senior Editor and Richard D Palmiter (Reviewer #1), who is a member of our Board of Reviewing Editors, Matthew Carter (Reviewer #2), and Chris Madden (Reviewer #3). The reviewers have discussed the reviews with one another and the Reviewing Editor has drafted this decision to help you prepare a revised submission.

Summary:

The manuscript clearly demonstrates that peripheral delivery of orexin either before or after initiation of sepsis by delivery of a lethal dose of lipopolysaccharide (LPS) can greatly ameliorate the devastating consequences. Furthermore, delivery of orexin to the CNS at a dose that is ineffective peripherally can also ameliorate the effects of LPS in this paradigm.

Essential revisions:

All three reviewers recommend that (1) either the serotonergic experiments be removed, or (2) developed more thoroughly (as described below). We recognize the desire to understand the mechanism by which orexin acting centrally might mitigate the effects of sepsis and the intriguing possibility that serotonin might be involved; however, the current experiments are incomplete and do not prove that they are relevant. Furthermore, we believe that it would require substantial effort to define the neural substrates of orexin in this context, especially if there are several. We believe that the experiments presented in the first part of the paper are sufficiently novel and important that they can stand on their own.

Hopefully the comments below will help you revise the manuscript.

*Reviewer #1:*

The authors of this paper provide a compelling set of experiments demonstrating that peripheral administration of orexin can ameliorate the consequences of sepsis induced by a lethal dose of lipopolysaccharide (LPS). They show that administration of orexin is beneficial even when administered after LPS, suggesting that it could have therapeutic applications. They demonstrate that orexin gains access to the brain due to breakdown of the blood-brain barrier during sepsis and that it acts there to mitigate the effects of LPS. They go on to show that one target of orexin action is the ventral raphe serotonergic neurons. Overall, this paper makes important contributions to understanding sepsis and provides a novel therapeutic avenue.

In their attempts to define the CNS targets of orexin action, the authors zero in on serotonergic neurons in the ventral raphe. They performed an experiment in which they inhibited ventral raphe serotonergic neurons and show that prevented the rise in body temperature normally observed with orexin treatment. The effect of orexin on body temperature is rather small – only 1 or 2 °C; thus, while statistically significant, it is not clear that is sufficient to prevent lethality. These experiments would be more compelling if the authors could show that orexin activation of serotonergic system contributes to survival after LPS.

*Reviewer #2:*

This manuscript studies the effects of peripheral and central injection of orexin on the inflammatory response that follows a relatively high dose of lipopolysaccharide (LPS). The authors show that peripherally administered orexin can increase survival following LPS injection, that it can inhibit cytokine production, and that it can keep body temperature and heart rate near normal levels. Importantly, they show that these effects are absent in orexin receptor double knockout mice. They further show that orexin can cross the blood brain barrier during endotoxin shock, and that orexin mediates increases in body temperature by activating the serotoninergic raphe nuclei.

This manuscript makes two fundamental and interesting discoveries: (1) peripherally administered orexin has therapeutic effects on endotoxin-mediated inflammation; (2) that peripherally administered orexin can cross the blood brain barrier under conditions of endotoxin shock and cause the activation of CNS systems. These results are novel and have therapeutic implications for peripherally-injected neuropeptides. The experiments are well executed and the results are convincing. My main concern stems from overall conclusions that don't seem justified by the results provided. I would suggest either adding further experiments to justify these conclusions, or revising the main conclusions of the paper (including the title) to better represent the scope of the data presented.

1) The major conclusion of this paper (i.e., the title), is that "peripherally administered orexin improves survival of mice with endotoxin shock by acting in the central nervous system." To me, the results of the manuscript are consistent with this statement but not conclusive. To show that orexin exerts protective effects by acting on the central nervous system, somehow the authors would need to block orexin signaling only in the brain. Because centrally injected orexin only recapitulated some (but not all) of the effects of peripherally injected orexin, it is possible that the central nervous system mediates some, but not necessarily all of these effects. I suggest refining the title because it is still unknown if orexin in the CNS specifically mediates survival (see comment #2 below).

2) The authors present a valid but precarious argument as to the mechanism by which orexin attenuates the inflammatory response: orexin crosses the blood brain barrier (Figure 3); orexin in the brain activates thermoregulatory neurons (Figure 5); survival in mice following endotoxin shock correlates with higher body temperatures (Figure 4); and inhibiting the raphe nuclei attenuates orexin-mediated stabilizing of body temperature. All of these experiments are well done, but the only conclusion that can be drawn is that "peripheral injection of orexin depends on the raphe nuclei to stabilize body temperature." Given the title of the paper, it would be excellent to repeat the experiments of Figure 5 (inhibiting the raphe nuclei during orexin injection) while performing a survival experiment. This experiment would clearly show that orexin enhances survival rates by raising body temperature, as is implied by the Results and Discussion sections.

3) The authors also suggest that orexin exerts anti-inflammatory effects by modulating levels of peripheral catecholamines or corticosterone. The authors could provide much more conclusive evidence for these claims by repeating these experiments in the presence of catecholamine antagonists or a corticosterone agonist. These experiments would demonstrate that modulation of catecholamines or corticosterone is necessary for the actions of orexin on endotoxin shock. Otherwise, the authors should be careful to note that the observed effects of orexin on catecholamines and corticosterone don't show necessity or sufficiency on these endocrine systems for orexin's actions.

*Reviewer #3:*

This manuscript explores the potential for peripheral administration of orexin in the treatment of endotoxin shock and the role of an action within the central nervous system in the therapeutic effect of this treatment. The main finding of this study is that peripheral administration of orexin increases survival in mice with endotoxin shock. Furthermore, data is presented demonstrating that during LPS-induced endotoxin shock, peripherally administered orexin crosses the blood brain barrier. In addition, ICV administration of orexin also increased survival of mice with endotoxin shock. Other important measures aimed at providing insight into the potential mechanism(s) by which orexin increases survival include cytokine levels (which are generally decreased by orexin administration), and body temperature (which is significantly decreased by LPS and partially rescued by orexin administration). While the overall findings of a new approach to treatment of endotoxin shock are of great interest, the experiments addressing the mechanism(s) by which this occurs are only observational and in general do not directly test hypotheses about the roles of the measured variables in mediating the increased survival. Nonetheless, these data should provide important bases for future studies addressing the mechanism(s) involved in the increased survival in mice treated with orexin during endotoxin shock.

---

## [Author Response]

*Essential revisions:*

*All three reviewers recommend that (1) either the serotonergic experiments be removed, or (2) developed more thoroughly (as described below). We recognize the desire to understand the mechanism by which orexin acting centrally might mitigate the effects of sepsis and the intriguing possibility that serotonin might be involved; however, the current experiments are incomplete and do not prove that they are relevant. Furthermore, we believe that it would require substantial effort to define the neural substrates of orexin in this context, especially if there are several. We believe that the experiments presented in the first part of the paper are sufficiently novel and important that they can stand on their own.*

We appreciate Dr. Westbrook’s thoughtful suggestions. Following his and reviewers’ recommendations, we have decided to delete the section describing the serotonergic DREADD experiments.

*Hopefully the comments below will help you revise the manuscript*

*Reviewer #1: […] In their attempts to define the CNS targets of orexin action, the authors zero in on serotonergic neurons in the ventral raphe. They performed an experiment in which they inhibited ventral raphe serotonergic neurons and show that prevented the rise in body temperature normally observed with orexin treatment. The effect of orexin on body temperature is rather small – only 1 or 2 °C; thus, while statistically significant, it is not clear that is sufficient to prevent lethality. These experiments would be more compelling if the authors could show that orexin activation of serotonergic system contributes to survival after LPS.*

We entirely agree with the reviewer that it would be ideal if we could positively demonstrate the role of serotonergic activation in the improved survival. However, the survival DREADD experiments are extremely challenging, because we have to be able to continuously inhibit serotonergic neurons by CNO at least for a few days (there is no precedence as far as we know), and we need very large amounts of orexin peptide for continuous infusion. While we are still considering the feasibility of such experiments for a future project, in this paper, we have deleted the section describing the serotonergic DREADD experiments, following the suggestion from the Senior Editor.

*Reviewer #2: […] This manuscript makes two fundamental and interesting discoveries: (1) peripherally administered orexin has therapeutic effects on endotoxin-mediated inflammation; (2) that peripherally administered orexin can cross the blood brain barrier under conditions of endotoxin shock and cause the activation of CNS systems. These results are novel and have therapeutic implications for peripherally-injected neuropeptides. The experiments are well executed and the results are convincing. My main concern stems from overall conclusions that don't seem justified by the results provided. I would suggest either adding further experiments to justify these conclusions, or revising the main conclusions of the paper (including the title) to better represent the scope of the data presented.*

We appreciate and agree with the reviewer’s comments. Accordingly, we have changed the title as follows: “Peripherally administered orexin improves survival of mice with endotoxin shock.” We have also deleted the section describing the serotonergic DREADD experiments, as suggested by the Senior Editor.

*1) The major conclusion of this paper (i.e., the title), is that "peripherally administered orexin improves survival of mice with endotoxin shock by acting in the central nervous system." To me, the results of the manuscript are consistent with this statement but not conclusive. To show that orexin exerts protective effects by acting on the central nervous system, somehow the authors would need to block orexin signaling only in the brain. Because centrally injected orexin only recapitulated some (but not all) of the effects of peripherally injected orexin, it is possible that the central nervous system mediates some, but not necessarily all of these effects. I suggest refining the title because it is still unknown if orexin in the CNS specifically mediates survival (see comment #2 below).*

We agree with the reviewer. Our experiments do not conclusively show that CNS mediate all of orexin’s effects on the survival. In order to reveal that orexin in the CNS specifically mediates the survival, we have to investigate, for example, the effects of orexin on the survival in CNS-specific orexin receptor KO mice. According to the reviewer’s suggestion, we have modified the title of the paper and have deleted “by activating the brainstem raphe nuclei” in the Abstract.

*2) The authors present a valid but precarious argument as to the mechanism by which orexin attenuates the inflammatory response: orexin crosses the blood brain barrier (Figure 3); orexin in the brain activates thermoregulatory neurons (Figure 5); survival in mice following endotoxin shock correlates with higher body temperatures (Figure 4); and inhibiting the raphe nuclei attenuates orexin-mediated stabilizing of body temperature. All of these experiments are well done, but the only conclusion that can be drawn is that "peripheral injection of orexin depends on the raphe nuclei to stabilize body temperature." Given the title of the paper, it would be excellent to repeat the experiments of Figure 5 (inhibiting the raphe nuclei during orexin injection) while performing a survival experiment. This experiment would clearly show that orexin enhances survival rates by raising body temperature, as is implied by the Results and Discussion sections.*

We agree with the reviewer’s comment. We only showed the possibility that raphe nuclei are a target region of orexin in endotoxin shock. To substantiate the possibility, we have to perform a survival experiment while inhibiting the raphe nuclei during orexin injection as the reviewer suggested. However, the survival DREADD experiments are extremely challenging, because we have to be able to continuously inhibit serotonergic neurons by CNO at least for a few days (there is no precedence as far as we know), and we need very large amounts of orexin peptide for continuous infusion. While we are still considering the feasibility of such experiments for a future project, in this paper, we have deleted the section describing the serotonergic DREADD experiments, following the suggestion from the Senior Editor.

*3) The authors also suggest that orexin exerts anti-inflammatory effects by modulating levels of peripheral catecholamines or corticosterone. The authors could provide much more conclusive evidence for these claims by repeating these experiments in the presence of catecholamine antagonists or a corticosterone agonist. These experiments would demonstrate that modulation of catecholamines or corticosterone is necessary for the actions of orexin on endotoxin shock. Otherwise, the authors should be careful to note that the observed effects of orexin on catecholamines and corticosterone don't show necessity or sufficiency on these endocrine systems for orexin's actions.*

We agree that the levels of catecholamines and corticosterone do not show necessity or sufficiency with regard to the actions of orexin on endotoxin shock. However, such experiments with agonists/antagonists may often be tricky. We added the following sentences: “Therefore, the increase of corticosterone levels in endotoxin shock may play a role in the anti-inflammatory effects of orexin, although further studies with glucocorticoid agonists/antagonists are necessary to substantiate this possibility”, and “Further studies with adrenergic antagonists would fully clarify the role of these neuroendocrine changes in the observed improvements in survival”.

*Reviewer #3:*

*This manuscript explores the potential for peripheral administration of orexin in the treatment of endotoxin shock and the role of an action within the central nervous system in the therapeutic effect of this treatment. The main finding of this study is that peripheral administration of orexin increases survival in mice with endotoxin shock. Furthermore, data is presented demonstrating that during LPS-induced endotoxin shock, peripherally administered orexin crosses the blood brain barrier. In addition, ICV administration of orexin also increased survival of mice with endotoxin shock. Other important measures aimed at providing insight into the potential mechanism(s) by which orexin increases survival include cytokine levels (which are generally decreased by orexin administration), and body temperature (which is significantly decreased by LPS and partially rescued by orexin administration). While the overall findings of a new approach to treatment of endotoxin shock are of great interest, the experiments addressing the mechanism(s) by which this occurs are only observational and in general do not directly test hypotheses about the roles of the measured variables in mediating the increased survival. Nonetheless, these data should provide important bases for future studies addressing the mechanism(s) involved in the increased survival in mice treated with orexin during endotoxin shock.*

We agree with the reviewer’s comment that further investigation is required for revealing the mechanism(s).